# Development of EST-SSR Markers Related to Polyphyllin Biosynthesis Reveals Genetic Diversity and Population Structure in *Paris polyphylla*

Xiaoyang Gao [1,†], Qixuan Su [1,2,†], Baolin Yao [1,2], Wenjing Yang [1,3], Weisi Ma [4], Bin Yang [4] and Changning Liu [1,5,6,*]

1   CAS Key Laboratory of Tropical Plant Resources and Sustainable Use, Xishuangbanna Tropical Botanical Garden, Chinese Academy of Sciences, Kunming 650223, China; gaoxiaoyang@xtbg.ac.cn (X.G.); suqixuan@mail.ustc.edu.cn (Q.S.); yblbio@mail.ustc.edu.cn (B.Y.); yangwenjing@xtbg.ac.cn (W.Y.)
2   School of Life Sciences, University of Science and Technology of China, Hefei 230026, China
3   College of Life Sciences, University of Chinese Academy of Sciences, Beijing 100049, China
4   Institute of Medicinal Plants, Yunnan Academy of Agricultural Sciences, Kunming 650205, China; masilaoq@aliyun.com (W.M.); yb@yaas.org.cn (B.Y.)
5   Center of Economic Botany, Core Botanical Gardens, Chinese Academy of Sciences, Xishuangbanna 666303, China
6   The Innovative Academy of Seed Design, Chinese Academy of Sciences, Kunming 650223, China
*   Correspondence: liuchangning@xtbg.ac.cn; Tel.: +86-0691-8715071
†   These authors contributed equally to this work.

**Abstract:** *Paris polyphylla* is an important medicinal plant that can biosynthesize polyphyllins with multiple effective therapies, ranging from anti-inflammation to antitumor; however, the genetic diversity of *Paris polyphylla* is still unclear. To explore the genetic characteristics of cultivation populations in primary planting areas, we developed 10 expressed sequence tag simple sequence repeat (EST-SSR) markers related to polyphyllin backbone biosynthesis and utilized them in 136 individuals from 10 cultivated populations of *P. polyphylla* var. *yunnanensis*. The genetic diversity index showed that ten loci had relatively high genetic polymorphism levels. Shannon information of loci suggested that more information occurred within population and less information occurred among population. In addition, the overall populations exhibited a low degree of differentiation among populations, but maintained a high degree of genetic diversity among individuals, resulting in high gene flow and general hybridization. The genetic structure analysis revealed that 10 populations possibly derived from two ancestral groups and all individuals were found with different levels of admixture. The two groups were different from the cultivation groups at population level, suggesting the cross-pollination among cultivars. These findings will provide insights into the genetic diversity of the germplasm resources and facilitate marker-assisted breeding for this medicinal herb.

**Keywords:** *Paris polyphylla*; transcriptome; polyphyllin; EST-SSR markers; genetic diversity





## 1. Introduction

*Paris polyphylla* Smith is an important medicinal perennial herb, mainly distributed from Southwest China to the pan-Himalayan region [1]. *P. polyphylla* is the most in demand of the genus, and has dominated the industrialization and utilization of medicinal plants in Southwest China. Due to the remarkable effects on hemostasis, anti-inflammation, and anti-cancer, its dried rhizome becomes the key raw material for about 80 kinds of famous patented medicines [2]. According to previous phytochemical studies, polyphyllins are regarded as the chief active ingredients in this plant, and 174 different polyphyllins have been identified so far, which account for 54% (323) of the total number of known bioactive compounds [3]. Notably, there has been a 300-fold increase in the market price paid for the rhizomes during the past nearly forty years and approximately 1000 t of the rhizomes are sold annually [4]. However, the scarcity of *P. polyphylla* becomes the

bottleneck of the related pharmaceutical industry in recent years, mainly because of its long growth cycles (ca. 7–10 years), long-term seed dormancy, and the sharp increase in the demand for rhizomes by the pharmaceutical industry. In addition, wild resources are threatened by long-term over-exploration and habitat fragmentation, and natural resources of *P. polyphylla* become increasingly endangered. Therefore, it is necessary to investigate the germplasm resources and understand the differentiation for genetic resource conservation and sustainable utilization.

The genetic information of plants plays an essential part in formulating conservation strategies. Molecular markers become useful tools to study the genetic diversity and population structure of germplasm resources in non-model plants with no reference genomes [5]. A variety of molecular markers, including amplified restriction fragment length polymorphism (ARFLP), random amplified of polymorphic DNA (RAPD), restriction site amplified polymorphism (RSAP), start codon-targeted polymorphism (SCoT), sequence-related amplified polymorphism (SRAP), inter simple sequence repeat (ISSR), and simple sequence repeat (SSR) have extensively been used in plant source conservation and genetic breeding [6]. Among the different classes of molecular markers, SSR markers have become a particularly important tool because they are co-dominant, polymorphic, low-cost, and high efficiency; the putative function of SSR markers can often be deduced by a homology search [7]. Compared to SSR, expressed sequence tag simple sequences repeat (EST-SSR) has the advantage of offering more transferability among plant species and is widely used in plant genetic mapping [8,9]. To date, there are only a few reports about the genetic diversity of *P. polyphylla* by the aforementioned molecular markers. Previously, genetic diversity of the three cultivated populations of *P. polyphylla* var. *yunnanensis* was slightly higher than that of the three wild populations (0.153 vs. 0.151) through ISSR markers, suggesting the introduction and artificial selection of cultivars from comparatively wide areas of origin and subsequent gene flow among populations in cultivation [10]. Only 1.35% of genetic variations existed between 15 wild and 17 cultivated populations of *P. polyphylla* var. *yunnanensis* using AFLP markers, which indicates that there is no obvious genetic differentiation between wild and cultivated populations as result of the relatively short history of the domestication of cultivated populations [11]. SCoTs and SRAPs were developed to investigate genetic diversity using 33 *P. polyphylla* samples in the Dabie Mountains, which found that the polymorphisms and marker efficiency of SCoTs were higher than those of SRAPs [12]. Nine random EST-SSRs were detected based on a root transcriptome of *P. polyphylla* var. *yunnanensis*. Maker efficiency was then validated in 55 samples [13]. However, the lack of marker information for molecular phylogeny and genetic structure limited *P. polyphylla* collection, conservation, and utilization. *P. polyphylla* and other *Paris* species possess giant genomes, and none of their complete genomes have been sequenced so far [14]. There are, as of yet, few available EST sequences of *Paris* L. in the GenBank database. In addition, few studies have explored SSRs related to polyphyllin biosynthesis based high quality transcriptome.

In the present study, we identified a large number of EST-SSRs based on the transcriptome sequencing data of 36 tissue samples from our previous studies. The aims of this study were to: (1) develop SSR markers related to polyphyllin biosynthesis and validate their polymorphism levels; (2) explore the genetic background between germplasms from the primary planting areas of *P. polyphylla*. This study will provide novel insights into the genetic diversity of the germplasms from the major planting areas of *P. polyphylla* and aid in the conservation and utilization of this important medicinal plant.

## 2. Materials and Methods

### 2.1. Plant Material and DNA Extraction

In this study, healthy whole plants of 7-year-old *P. polyphylla* var. *yunnanensis* during the fruiting stage were widely sampled from the main grow areas in Yunnan Province, Southwest China. A total of 136 individuals from 10 populations of *P. polyphylla* var. *yunnanensis* were collected from major production areas, and all individuals were transplanted in

the green house of Xishuangbanna Tropical Botanical Garden, Chinese Academy of Sciences (Kunming, China). The samples collected covered the central, northwest, southwest, and west of Yunnan (Figure 1 and Table 1), and the samples collected included two kinds of widely grown varieties, namely short-stalked variety and long-stalked variety. The two varieties have obvious differences in stalk length, size of rhizome and leaf, fruit yield, etc. The characteristics of stalk are usually used to distinguish the two varieties. The detailed information for each population is shown in Figure 1 and Table 1. The fresh leaves were sampled and stored at −80 °C. The genomic DNA was extracted using CTAB method [15]. The DNA concentration was estimated with a NanoDrop-1000 spectrophotometer (Nano Drop Technologies, Wilmington, DE, USA) and normalized to 30 ng/μL for polymerase chain reaction (PCR).

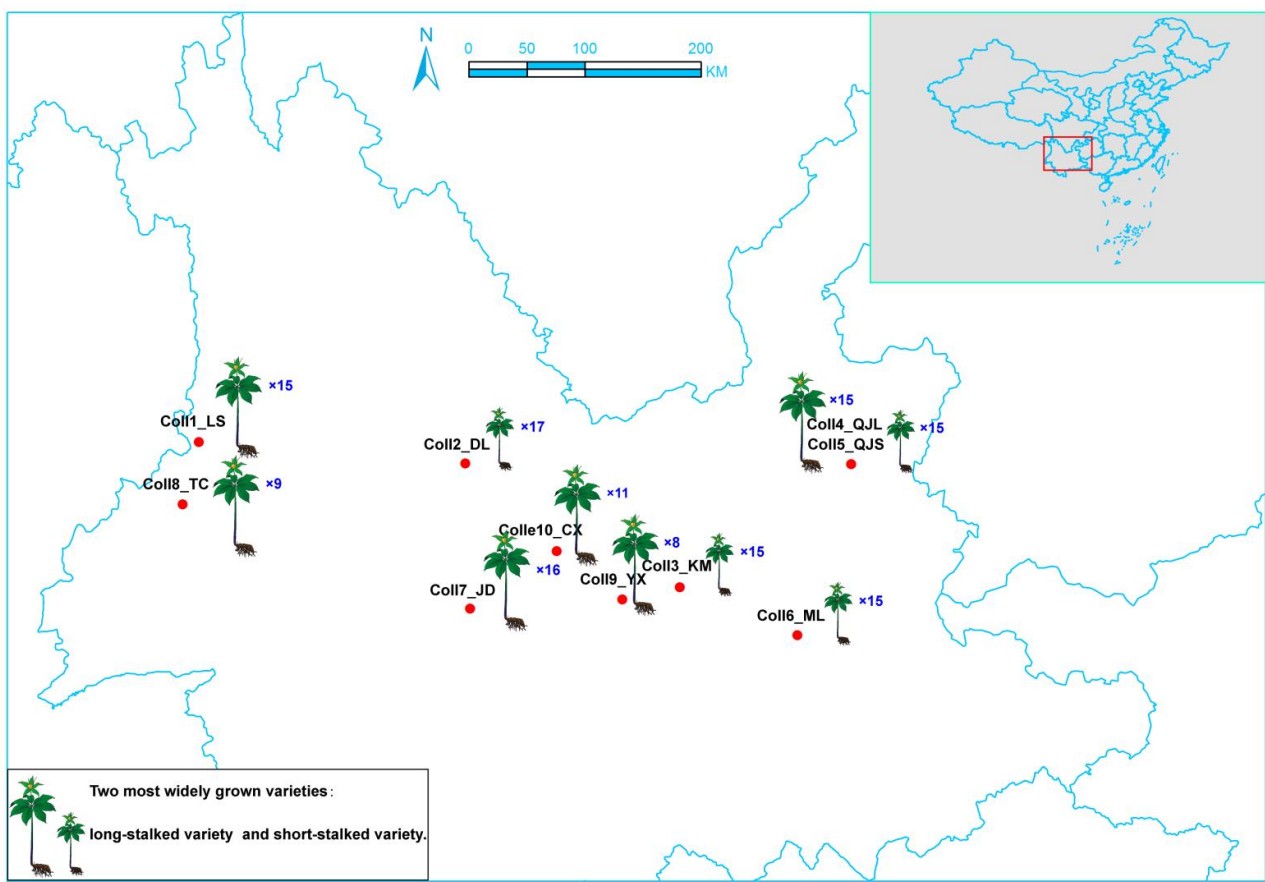

**Figure 1.** Sampling locations of 10 populations of *P. polyphylla* var. *yunnanensis* in the present study. The varieties with different stem heights represent the long-stalked variety and the short-stalked variety, respectively. The population size is also denoted.

**Table 1.** Sampling locations and number of samples analyzed in the present study.

| Population | Sampling Location | City | Population Size | Varieties | Longitude (°) | Latitude (°) | Altitude (m) |
|---|---|---|---|---|---|---|---|
| Coll1_LS | Lushui | Nujiang | 15 | long stalk | 98.78 (E) | 25.91 (N) | 1446 |
| Coll2_DL | Xiangyun | Dali | 17 | short stalk | 100.84 (E) | 25.74 (N) | 1775 |
| Coll3_KM | Xishan | Kunming | 15 | short stalk | 102.50 (E) | 24.78 (N) | 2008 |
| Coll4_QJL | Zhanyi | Qujing | 15 | long stalk | 103.83 (E) | 25.73 (N) | 2204 |
| Coll5_QJS | Zhanyi | Qujing | 15 | short stalk | 103.83 (E) | 25.73 (N) | 2204 |
| Coll6_ML | Mile | Honghe | 15 | short stalk | 103.41(E) | 24.41 (N) | 1711 |
| Coll7_JD | Jingdong | Puer | 16 | long stalk | 100.88 (E) | 24.62 (N) | 2237 |
| Coll8_TC | Tengchong | Baoshan | 9 | short stalk | 98.65(E) | 25.42 (N) | 1850 |
| Coll9_YX | Yimen | Yuxi | 8 | long stalk | 102.06 (E) | 24.69 (N) | 1873 |
| Coll10_CX | Chuxiong | Chuxiong | 11 | long stalk | 101.49 (E) | 24.95 (N) | 1857 |

### 2.2. EST-SSR Identification and Marker Development

All types of SSRs were identified through transcriptome analysis of *P. polyphylla* var. *yunnanensis*. A total of 36 tissues transcriptome sequencing data were from our previous study (12 tissue samples) [16] and subsequent transcriptome sequencing data (24 tissues samples, unpublished). The data are available in NCBI (PRJNA682903 and PRJNA630028). SSRs were ascertained using the microsatellite identification tool MISA 1.0 (Thiel Thomas, Seeland, Germany) [17]. The SSRs were considered to contain mono-, di-, tri-, tetra-, penta-, and hexa-nucleotides with minimum repeat numbers of 10, 5, 4, 3, 3, and 3, respectively. The distance between adjacent SSRs $\leq$ 100 bp was defined as compound SSR. The functional annotation of the gene contained SSRs were obtained through homology by searching against the public database GO, KEGG, Swiss-Prot, and Pfam using BLAST with an E-value cutoff of $10^{-5}$. The non-redundancy gene sequences associated with polyphyllin biosynthesis contained SSRs were filtered for primer design by Primer 3.0 (Andreas Untergasser, Heidelberg, Germany) [18]. Flank sequence length of SSR < 20 bp and sequence contained mononucleotide repeats were removed according to SSR locus. The primer length ranged 16–26 bp, production of PCR was 100–450 bp, optimum Tm was 55–57 °C, GC content was 40–60%, and oligonucleotides were synthesized at Shanghai Sangon Biological Engineering Technology (Shanghai, China).

### 2.3. Marker Validation

SSR-PCR amplification for all designed markers was initially carried out using 30 random individuals from 10 populations. The PCR reaction system: 2.5 µL 10× Tag buffer, 2 µL of 2.5 mmol·L$^{-1}$ dNTPs, 1 µL of 10 µmol·L$^{-1}$ for each of forward and reverse primer of DNA, 0.25 µL 2.5 Taq Plus DNA polymerase, 1 µL DNA template. The PCR reaction conditions and procedures were performed as follows: initial denaturation step of 95 °C for 3 min, followed by denaturation at 94 °C for 30 s, annealing temperature at 60 °C for 30 s, extension at 72 °C for 30 s, 10 cycles; denaturation at 94 °C for 30 s, annealing temperature at 55 °C for 30 s, extension at 72 °C for 30 s, 10 cycles. The final extension was performed at 72 °C for 7 min. the amplified PCR products were detected by 8% non-denaturing polyacrylamide gels and stained by nucleic acid dye. The selected PCR products labelled with TAMRA, FAM, and HEX were pooled before separation in ABI 3730XL (Applied Biosystems). The PCR products were separated using a 96-capillary 3730XL DNA Analyzer (Thermo Fisher Scientific, Waltham, MA, USA) and the peak patterns were sized by Genemapper 4.0 (Thermo Fisher Scientific, Waltham, MA, USA). The primer pairs and marker were evaluated and determined, which yielded clear, reproducible, and polymorphic bands with an expected size and clear fluorescence signal that were selected for subsequent allele identification of all individuals.

### 2.4. Data Analysis

The artificial proofreading for raw data was implemented by checking the capillary electrophoresis (CE) peak diagram. The bands with the same base size were represented by a similar peak at the same locus. The EST-SSRs were tested for selective neutrality by means of an $F_{\mathbf{ST}}$ outlier method using LOSITAN [19,20]. After the preliminary runs to estimate the mean neutral $F_{\mathbf{ST}}$, 20,000 simulations with the infinite allele model (IAM) were performed, according to parameter settings set by Ohtani et al. [21]. Outlier loci under positive or balancing selection were determined based on 99.5% confidence intervals. The clustering pattern of individuals and populations were revealed by neutral loci. The loci with $F_{\mathbf{ST}}$ outliers were excluded from the following analyses. The EST-SSR loci data was formatted for subsequent analyses using GenAlEx 6.5b2 (Peakall Rod, Canberra, Australia) [22]. The number of observed alleles (*Na*), the number of efficient alleles (*Ne*), the observed heterozygosity (*Ho*), the expected heterozygosity (*He*), *Nei*'s genetic diversity index (*h*), Shannon diversity index (*I*), polymorphic information content (PIC), major allele frequency (MP), etc., were calculated using POPGENE 1.32 (Naoko Takezaki, Kagawa, Japan) and Powermaker 3.25 (Kejun Liu, Raleigh, USA) [23,24]. The genetic differentiation coefficient

among populations ($F_{ST}$), intraspecific inbreeding coefficient ($F_{IS}$), population inbreeding coefficient ($F_{IT}$), and gene flow ($Nm$) were calculated using Arlequin 3.5 (Laurent Excoffier, Lausanne, Switzerland) [25]. $Nm$ was calculated followed $Nm = 0.25(1 - F_{ST})/F_{ST}$ [26]. The Hardy–Weinberg equilibrium (HWE) with the chi-squared test for each population and loci was analyzed using POPGENE 1.32 (Naoko Takezaki, Kagawa, Japan), which was adjusted using Bonferroni for multiple tests [27]. The principal coordinate analysis (PCoA) via covariance matrix with data standardization was conducted using GenAlEx 6.50b (Rod Peakall, Canberra, Australia) [22].

The *Nei*'s (1983) standard genetic distance among populations, individuals, and clustering trees based on the unweighted pair group method with arithmetic means (UPGMA) algorithm (bootstrap: 1000) were calculated and analyzed using PowerMarker 3.25 (Kejun Liu, Raleigh, NC, USA) [24]. The consensus tree was generated, edited, and visualized using Phylip 3.68 (Jacques D. Retief, Totowa, NJ, USA), MEGA 5.10 (Sudhir Kumar, Tokyo, Japan), and FigTree 1.4.2 (A. Rambaut, Edinburgh, UK), respectively [28–30]. The population genetic structure was determined by utilizing a Bayesian clustering analysis using SRUCTURE 2.3.4 (K Pritchard Jonathan, Oxford, UK) [31]. A total of ten independent simulations for each $K$ ranging from 1 to 10 were performed with a burn-in period of 100,000 steps followed by 100,000 Markov Chain Monte Carlo (MCMC) iterations using the Admixture Model. The most probable number of population groups ($K$) was determined with delta $K(\Delta K)$ through web-based STRUCTURE HARVESTER [32,33]. Repeated sampling analysis and genetic structural plots were analyzed by CLUMPP 1.1.2 (Mattias Jakobsson, Ann Arbor, MI, USA) and visualized by DISTRUCT 1.1 (Noah A. Rosenberg, Los Angeles, CA, USA) [34,35].

## 3. Results

### 3.1. EST-SSR Identification

Novel EST-SSR markers were developed based on the transcriptome assembled from different tissues during developmental stages [16]. In total, 102,472 different EST-SSRs were identified from 341,191 unigenes, distributed in 73,770 sequences, with an average of 0.22 SSR per unigene and a distribution density of one SSR per 2.61 kb. The repeated motifs of SSRs were diverse; mononucleotide (34.32%) and dinucleotide (37.91%) were the most common repeated motif types. Among these, A/T repeat motif was the most abundant type (91.74% of total mononucleotide repeats), followed by AG(24.66%)/CT(15.45%)/TC(18.17%) (Supplementary Material Figure S1). The compound SSRs were also identified, and the number of this type of SSR was approximately 13,630. The copy number of repeat motifs was unevenly distributed in different unit types. The most frequent copy number of mononucleotide repeats was 91, whereas the most frequent copy number of hexanucleotide was 16. The copy number of repeat motifs significantly decreased with the increasing length of repeat unit, particularly from dinucleotide to hexanucleotide. The genes that contained SSRs were functionally annotated by the public database GO, SwissProt, Pfam, and KEGG. The annotation result showed that 12,723 unigenes containing SSRs have similarities to the homologs of GO terms. 4568 were by KEGG pathways, 8365 were by Swiss-Prot, and 7950 were by Pfam domains. A total of 244 unigenes related to polyphyllin biosynthesis were identified after length filtering and redundancy removing. Among these, 64 unigenes contained different SSRs. Similar to the distribution of SSR motif types in the transcriptome, mononucleotide and dinucleotide accounted for the larger proportion (55%), which was followed by compound SSRs, accounting for 15% (Figure 2).

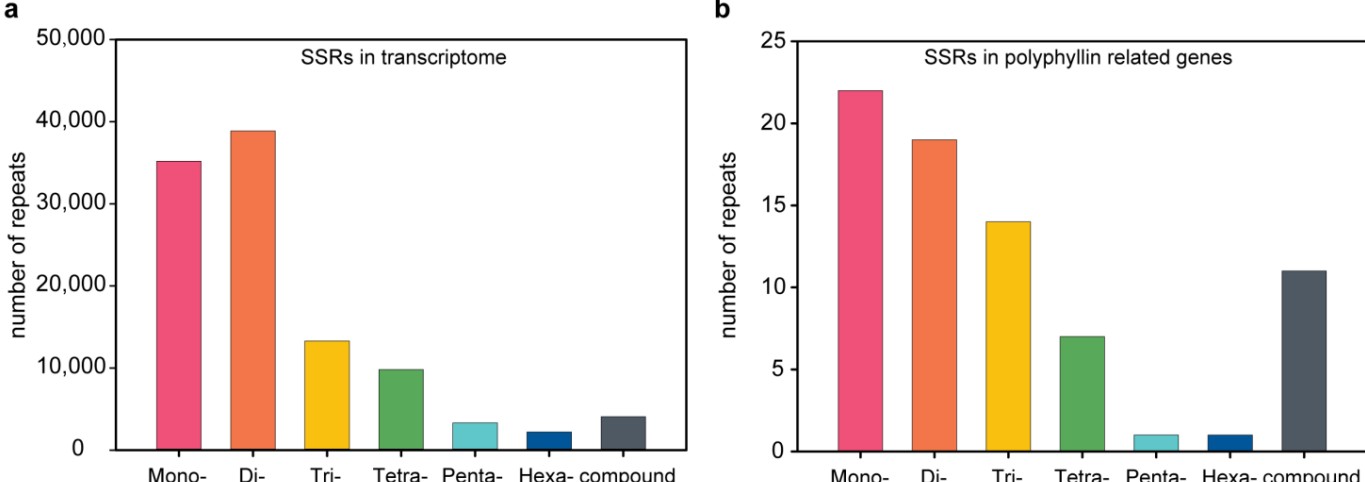

**Figure 2.** Statistics of EST-SSR based on the transcriptomic data. (**a**) number of different SSR types identified in the transcriptome: Mono-, Di-, Tri-, Tetra-, Penta-, Hexa-, and compound represent mononucleotide, dinucleotide, trinucleotide, pentanucleotide, hexanucleotide, and compound nucleotide, respectively; (**b**) number of different SSR related to polyphyllin biosynthesis.

### 3.2. Development of EST-SSR Makers

A total of 34 SSR primers related to polyphyllin biosynthetic genes were designed based on the conserved sequences at the 5′ or 3′ ends of SSR, taking into account of Tm values, hairpin structure, length of PCR product, etc. 34 pairs of SSR-PCR primers were firstly validated in 30 samples in the preliminary experiment, and their PCR products were evaluated by the results of agarose gel electrophoresis and capillary electrophoresis (CE). Finally, 12 primer pairs (35%) were determined and were applied in SSR-PCR and the genotype analysis of 136 samples (Table 2). All of the products successfully yielded clear, reproducible, and polymorphic bands with an expected size and clear fluorescence signal. As shown in Table 2, products from 6 primer pairs contained dinucleotide repeats, products from 3 primer pairs contained trinucleotide repeats, products from one primer pair contained tetranucleotide repeats, products from 2 primer pairs contained compound nucleotide repeats.

**Table 2.** Primers associated with polyphyllin biosynthetic gene designed in this study.

| Primers | Sequence (5′ to 3′) | Tm (°C) | SSR Type | Expected Product Size (bp) | 5′Modification | Gene Candidate | Polyphyllin Backbone Biosynthesis |
|---|---|---|---|---|---|---|---|
| STR1035-9F | CTATCGGAGAGTCTGACCCTAC | 55 | (GT)6 | 130 | 5′HEX | *STE24* | downstream |
| STR1035-9R | GTAACCATTGATTTCCAGCTG | | | | | | |
| STR1035-11F | CAGAATAAAGACGGTGAATTAAAAT | 56 | (CGC)4 | 115 | 5′HEX | *SMT2* | downstream |
| STR1035-11R | CCCATGCATATGATCCTCTG | | | | | | |
| STR1035-13F | AAGCTGGAATCAACCATAAACT | 55 | (AG)5 | 124 | 5′HEX | *SQLE* | downstream |
| STR1035-13R | AGAGCAGGAGAAACCCTAGAA | | | | | | |
| STR1035-14F | TGCTAAAAAGGCTGGTGATATC | 57 | (AG)11*(A)10 | 111 | 5′HEX | *DXS* | upstream |
| STR1035-14R | CGGCTTTCACTGTTTCACATA | | | | | | |
| STR1035-15F | CAAATAATATGATCCCTACAGAAGA | 56 | (TTA)4 | 191 | 5′6-FAM | *HMGS* | upstream |
| STR1035-15R | TAATAATAGCAGTTCCACATTCAGT | | | | | | |
| STR1035-18F | GCAGAAACTGTACCATGAGGAG | 57 | (CAAA)3 | 268 | 5′6-FAM | *FNTA* | downstream |
| STR1035-18R | CGTCTTGCTTGATTAACTAGGATT | | | | | | |
| STR1035-22F | CGATCCGAATCCTCTGTTAAA | 56 | (CT)5 | 191 | 5′6-FAM | *MVD* | upstream |
| STR1035-22R | GTCACCATTAGGATCCATTTCT | | | | | | |
| STR1150-1F | CAAGCTATTCGCCGTCCT | 56 | (CGC)4*(ACG)4 | 427 | 5′6-FAM | *HMGR* | upstream |
| STR1150-1R | CTGCCCCAGAATCGAGC | | | | | | |
| STR1150-3F | ATCTCCACGCCTTCCCTT | 57 | (CCA)4 | 170 | 5′6-FAM | *ispD* | upstream |
| STR1150-3R | CTCTGCTTCTCTTTTCGCAAT | | | | | | |
| STR1150-4F | AGGATAACTAACAAAAGAGAGGATG | 56 | (TC)5 | 190 | 5′6-FAM | *ispE* | upstream |
| STR1150-4R | TCTTCCTATAGAGGTTGAGTGCT | | | | | | |
| STR1150-7F | TGCCCCCCCTCATCTC | 56 | (TC)5 | 140 | 5′6-FAM | *TGL4* | downstream |
| STR1150-7R | GGAAATTCTTGAGCTTGCAGT | | | | | | |
| STR1150-9F | GTGCCCGTTCCATTCAAG | 57 | (GA)10 | 119 | 5′6-FAM | *MVK* | upstream |
| STR1150-9R | TGCTCGCCGGAGAGTATG | | | | | | |

### 3.3. Polymorphism Analysis of SSR Loci

The EST-SSR markers in this study were developed based on expressed sequence tags derived from the transcriptome data. Firstly, the SSR loci were tested for selective neutrality, and the SSRs with $F_{ST}$ outliers were filtered from the following analyses. The LOSITAN analysis detected two $F_{ST}$ outliers, indicating that the loci 1035P11 and 1035P14 were probably under positive selection (Figure 3); these two outlier loci were excluded from all subsequent analyses; the remaining loci under neutral selection were reserved for the subsequent genetic variation analyses.

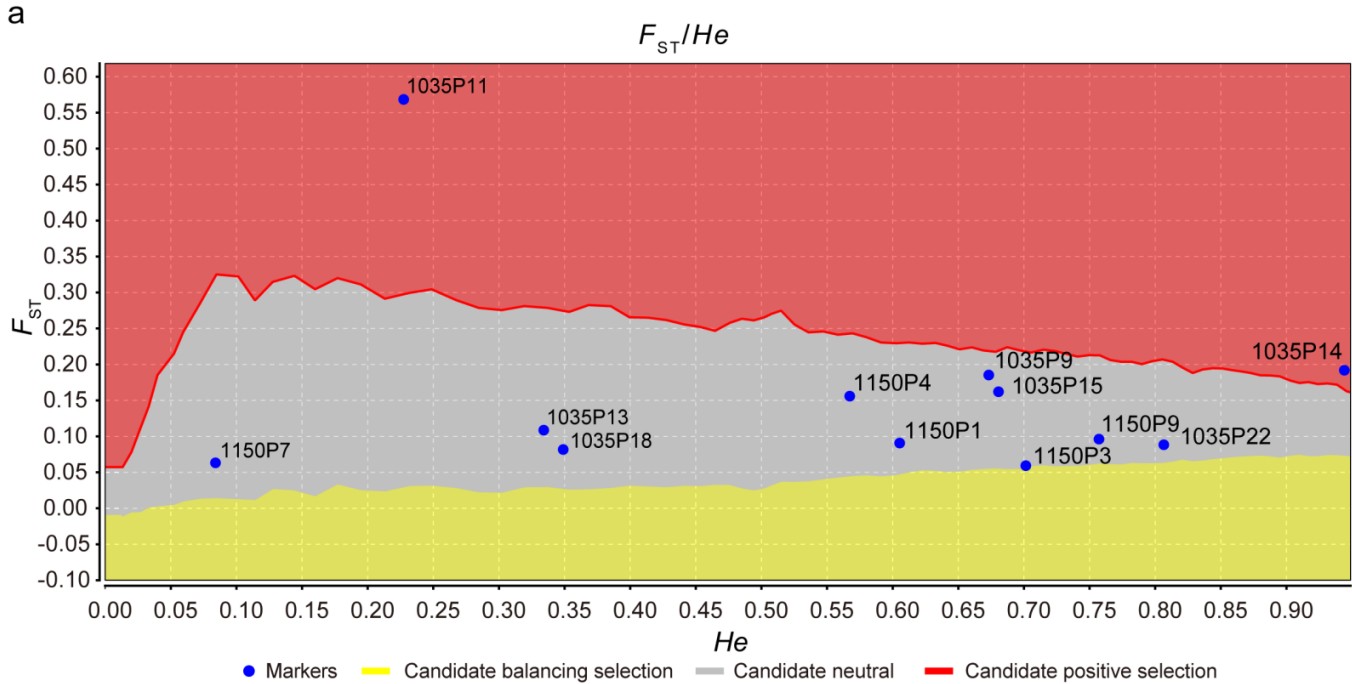

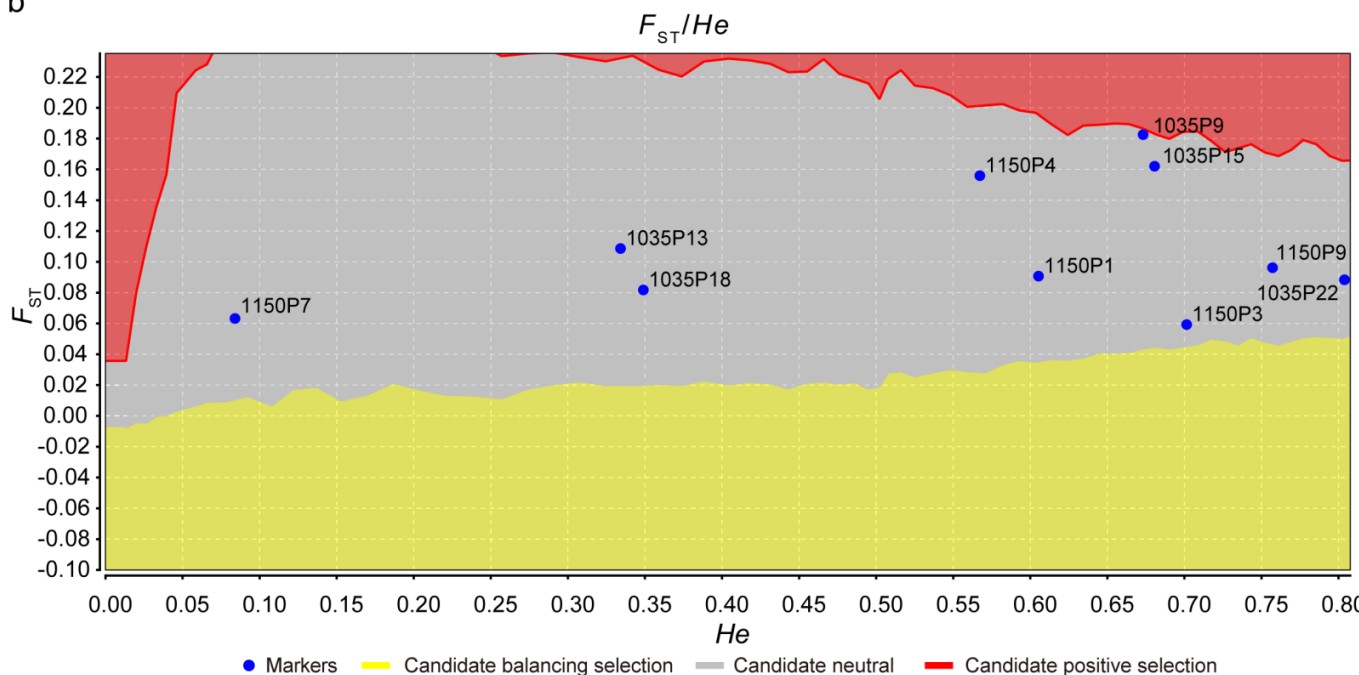

**Figure 3.** Assessment of $F_{ST}$ outlier EST−SSR loci and neutrality tests. (**a**) analysis of $F_{ST}$ outliers of 12 SSR loci; (**b**) neutrality tests for 10 SSR loci.

The genetic variation analysis of these loci was implemented based on 136 individuals from 10 populations. As shown in Table 3, *Na* was 99 alleles in total, and it ranged from 5 to 13, with an average of 9.90 alleles per locus. *Ne* was 27.92 in total, ranging from 1.1018 to 4.9541. *Ho* ranged from 0.3088 to 0.9599. *He* ranged from 0.0927 to 0.8011. *h* ranged from 0.0924 to 0.7981. PIC can detect and reflect the genetic variation level [36]. PIC value is grouped into highly polymorphic (PIC > 0.5), moderate polymorphic (0.5 > PIC > 0.25), and low polymorphic (PIC < 0.25) categories. PIC of 10 SSR loci ranged from 0.0842 to 0.7684. According to the Bostein theory, PIC values of 9 loci were over 0.25 and they were relatively high polymorphic loci, except locus 1150P7. Locus 1035P22 has the highest PIC value (0.7684), and the Shannon's information index of this locus is 1.8423. PIC values of the loci were consistent with *He* values of the corresponding loci. In general, the average *Ne*, *He*, and PIC were 2.7920, 0.5600, 0.5225, indicating that 10 screened EST-SSR loci had relatively high genetic polymorphism levels. Among the 10 loci, 1150P7 had the lowest level of genetic diversity and 1035P22 had the highest level of genetic diversity.

**Table 3.** The genetic diversity and genetic variation of EST-SSR loci. ** represent deviating from Hardy-Weinberg equilibrium (HWE) at the level of $p < 0.01$.

| Loci | Na | Ne | I | PIC | Ho | He | Nei (h) | HWE | $F_{IS}$ | $F_{ST}$ | $F_{IT}$ | Nm |
|---|---|---|---|---|---|---|---|---|---|---|---|---|
| 1035P9 | 9 | 3.0711 | 1.3396 | 0.6202 | 0.5515 | 0.6769 | 0.6744 | 0.11 | −0.0311 | 0.1983 | 0.1734 | 1.0105 |
| 1035P13 | 5 | 1.5449 | 0.5833 | 0.3102 | 0.2132 | 0.354 | 0.3527 | ** | 0.3055 | 0.1281 | 0.3945 | 1.7010 |
| 1035P15 | 7 | 3.1617 | 1.3559 | 0.6505 | 0.1544 | 0.6862 | 0.6837 | ** | 0.7419 | 0.1781 | 0.7879 | 1.1533 |
| 1035P18 | 11 | 1.4961 | 0.821 | 0.3207 | 0.1103 | 0.3328 | 0.3316 | ** | 0.5952 | 0.1091 | 0.6393 | 2.0421 |
| 1035P22 | 13 | 4.9541 | 1.8423 | 0.7684 | 0.3235 | 0.8011 | 0.7981 | ** | 0.5045 | 0.1131 | 0.5605 | 1.9598 |
| 1150P1 | 12 | 2.5313 | 1.4029 | 0.5785 | 0.1838 | 0.6072 | 0.6049 | ** | 0.6646 | 0.1145 | 0.7030 | 1.9332 |
| 1150P3 | 12 | 3.3544 | 1.4452 | 0.6559 | 0.5368 | 0.7045 | 0.7019 | ** | 0.1581 | 0.0866 | 0.2310 | 2.6380 |
| 1150P4 | 5 | 2.3126 | 0.9829 | 0.4781 | 0.1691 | 0.5697 | 0.5676 | ** | 0.6610 | 0.1720 | 0.7193 | 1.2035 |
| 1150P7 | 5 | 1.1018 | 0.2441 | 0.0842 | 0.0441 | 0.0927 | 0.0924 | ** | 0.4801 | 0.0855 | 0.5246 | 2.6727 |
| 1150P9 | 20 | 4.3902 | 2.1041 | 0.7585 | 0.6912 | 0.7751 | 0.7722 | 0.02 | −0.0132 | 0.1182 | 0.1065 | 1.8653 |

F-statistic estimates for 10 SSR loci of 10 populations showed the genetic differentiation and inbreeding coefficients. $F_{ST}$ values for these loci were estimated 0.0855–0.1983, with an average of 0.1304, which indicated that there was little genetic variation among the populations and 86.96% of the genetic variation was within populations. The low $F_{ST}$ implied a low level of differentiation among the populations of *P. polyphylla* var. *yunnanensis*. $F_{IS}$ values of 5 loci were over 0.50, and the high $F_{IS}$ implied a considerable degree of inbreeding. Whereas, $F_{IS}$ values of locus 1035P9 and 1150P9 showed the excess heterozygosity with negative $F_{IS}$ values (−0.0311, −0.0312), suggesting outbreeding. $F_{IT}$ values ranged from 0.1065 to 0.7879, with an average value of 0.4840. In this study, *Nm* values were estimated to be ranged from 1.0105 to 2.6727. The average *Nm* value of the loci was 1.82 (*Nm* > 1). Two loci (locus 1035P9 and locus 1150P9) were in accordance with the Hardy–Weinberg equilibrium (HWE), but eight loci showed significant departures from HWE after Bonferroni correction, apparently due to heterozygote deficiency. According to the Shannon informational diversity statistics portioned by population and total for codominant data averaged across loci, 25% of total information occurred among populations, 75% of total information occurred within population (Figure S2a).

## 3.4. Genetic Diversity and Genetic Variation of Populations

The genetic diversity parameters of 10 populations were also estimated and the populations displayed abundant genetic diversity (Table 4). The *Na* ranged from 2.4 to 4.9 with an average value of 3.79. *Ne* ranged from 1.69 to 2.78 with an average value of 2.32. *Ho* value ranged from 0.2182 to 0.3800 with average value of 0.29. Population QJL, QJS, and LS had high *Ho* values. *He* values ranged from 0.35 to 0.57 with average value of 0.49. Population QJL and QJS had high *He* values. Shannon diversity index (*I*) ranged from 0.53 to 1.09, and the maximum and the minimum of *I* were from population QJS and

TC, respectively. The values of *h* can reflect the population variations, which ranged from 0.33 to 0.56 with average value of 0.47. The *h* values and the ranks were in accordance with results of *He* values. Populations QJL, QJS, and LS had higher genetic diversity ($h > 0.50$, $Ne > 2.3$), the population from TC had lowest genetic diversity ($h = 0.33$, $Ne = 1.69$). Based on the above, population QJS, LS had higher genetic diversity than other populations, whereas population TC had the lowest genetic diversity. AMOVA was carried out to assess the overall distribution of diversity within and among populations (Figure S2b). Of the total genetic diversity, 1% of the variation occurred among populations, 69% occurred among individuals, and 30% occurred within individual; thus, AMOVA supported the results of *Nei*'s genetic statistics and the Shannon diversity estimation that there is a low degree of population differentiation.

**Table 4.** The genetic diversity and genetic differentiation of cultivated populations.

| Pop | Major Allele Frequency | Genotype Number | *Na* | *Ne* | Gene Diversity | PIC | *I* | *Ho* | *He* | *Nei* (*h*) |
|---|---|---|---|---|---|---|---|---|---|---|
| LS | 0.58 | 4.9 | 4.4 | 2.7809 | 0.5531 | 0.5078 | 1.0689 | 0.3000 | 0.5685 | 0.5496 |
| DL | 0.61 | 4.1 | 3.2 | 2.2729 | 0.4635 | 0.4124 | 0.8135 | 0.2353 | 0.4718 | 0.4580 |
| KM | 0.66 | 5.3 | 4.7 | 2.4983 | 0.4593 | 0.4315 | 0.9585 | 0.3200 | 0.4759 | 0.4600 |
| QJL | 0.56 | 5.5 | 4.9 | 2.7692 | 0.5553 | 0.5073 | 1.0883 | 0.3733 | 0.5740 | 0.5549 |
| QJS | 0.59 | 6.0 | 5.0 | 2.4917 | 0.5489 | 0.5092 | 1.0906 | 0.3800 | 0.5678 | 0.5489 |
| ML | 0.58 | 4.3 | 3.4 | 2.3365 | 0.5087 | 0.4419 | 0.8852 | 0.3133 | 0.5262 | 0.5087 |
| JD | 0.63 | 4.9 | 4.4 | 2.2769 | 0.4801 | 0.4343 | 0.9317 | 0.2687 | 0.5016 | 0.4859 |
| TC | 0.75 | 2.3 | 2.2 | 1.6859 | 0.3278 | 0.2782 | 0.5346 | 0.2889 | 0.3471 | 0.3278 |
| YX | 0.69 | 2.6 | 2.4 | 1.9034 | 0.4094 | 0.3509 | 0.671 | 0.2375 | 0.4367 | 0.4094 |
| CX | 0.67 | 4.0 | 3.3 | 2.1356 | 0.4343 | 0.3905 | 0.7938 | 0.2182 | 0.4550 | 0.4343 |

*3.5. Genetic Structure and Population Clustering*

To infer the genetic structure, the coancestry relationships of the populations were analyzed based on a Bayesian module using the STRUCTURE program. The results showed that when $K = 2$, $\Delta K$ reached the maximum value, which indicates that 10 populations mainly came from two ancestral groups (Figure 4a,b). The individuals in the groups were composed with admixed populations. In addition, UPGMA clustering of populations was constructed based on *Nei*'s (1983) genetic distance among populations. The unrooted tree of populations revealed that the populations were clustered into two groups, which was broadly consistent with the genetic population structure results (Figure 4c). Following the same analysis procedure, the tree of individuals was conducted subsequently (Figure 5). The clustering results showed that individuals of different populations clustered together and the populations with higher genetic diversity index were inclined to have individuals admixing with other populations. The PCoA results showed that the first and second principal components accounted for 36.8% and 33.0% of the total genetic variation, respectively (Figure 4d); it also showed that populations from two cultivated groups got close to each other, such as population JD and QJS. Most individuals were roughly clustered according to their corresponding populations (Figure S3).

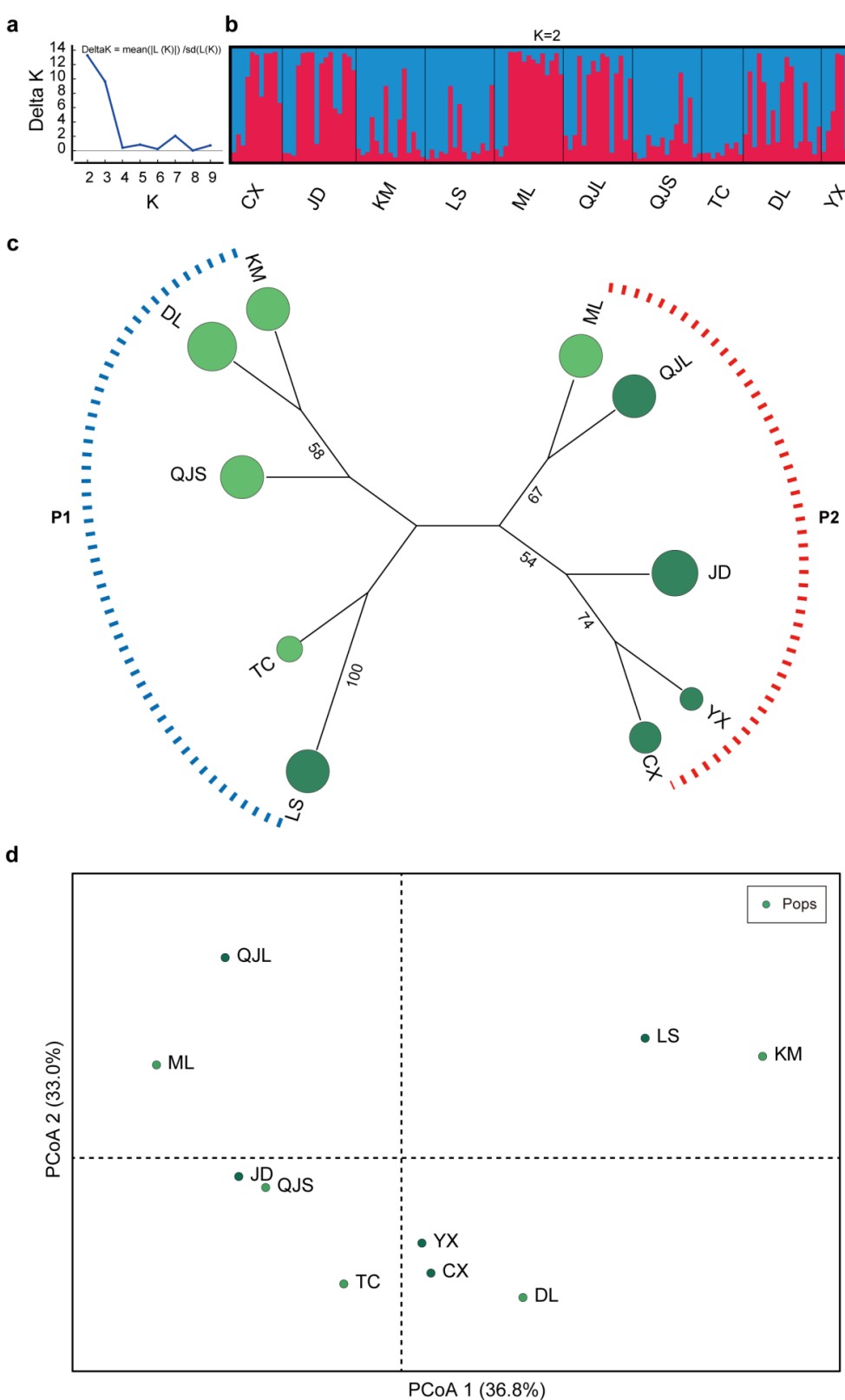

**Figure 4.** The population genetic structure of *P. polyphylla* var. *yunnanensis*. (**a**) relationships between the number of clusters (*K*) and the corresponding log probability of the data L(Δ*K*); (**b**) assignment of individuals to *K* = 2 genetically distinguishable group. (**c**) genetic divergence of 10 populations based on UPGMA cluster analysis. The two cultivated populations are denoted with dark green and light green. (**d**) principal coordinate analysis (PCoA) of 10 populations.

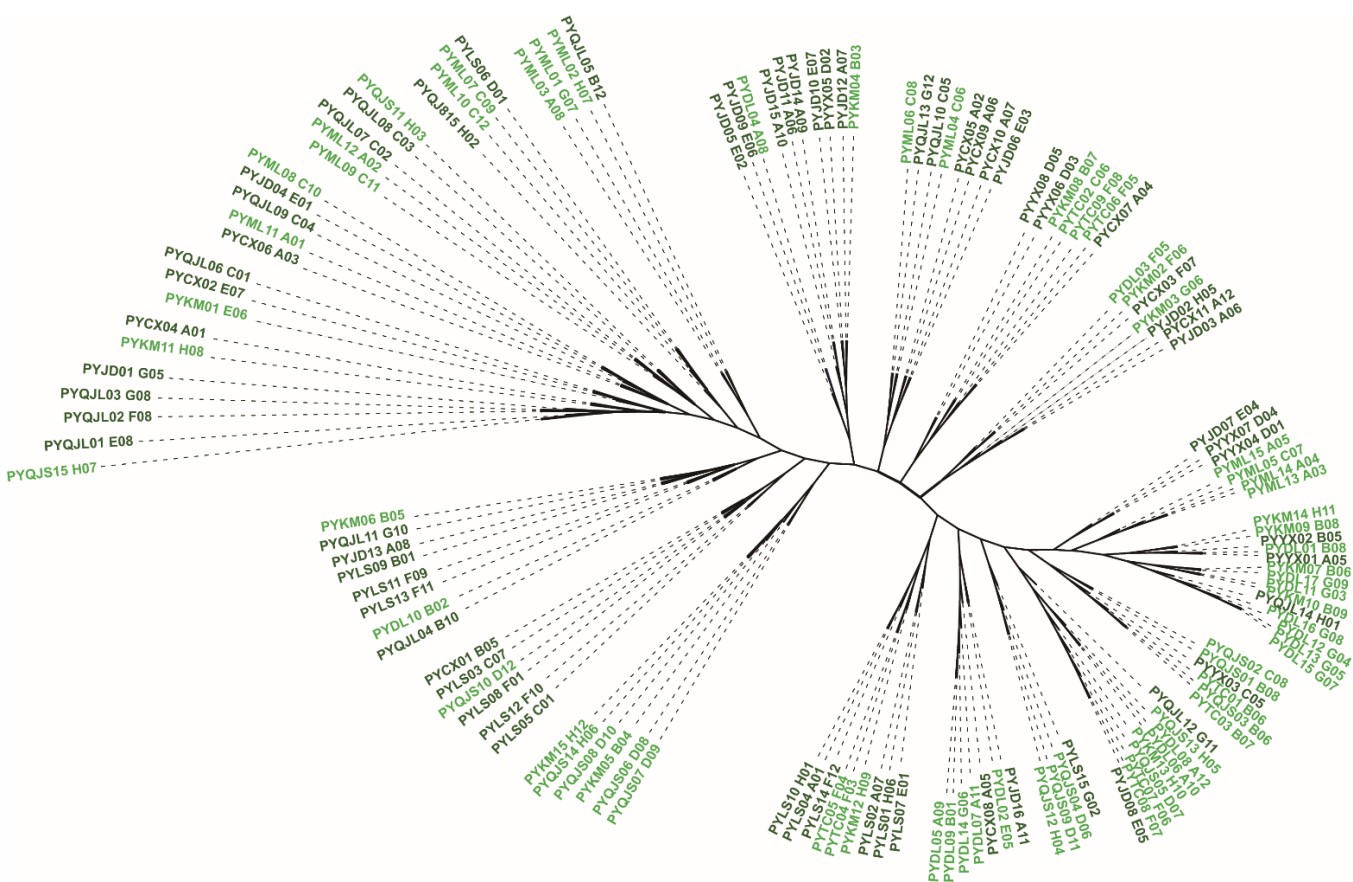

**Figure 5.** Genetic divergence of 136 individuals based on UPGMA cluster analysis. The long-stalked variety and the short-stalked variety are denoted with dark green and light green.

## 4. Discussion

### 4.1. SSR Frequency and Distribution

SSR holds a great promise for exploiting genetic diversity, characterizing accumulated phenotypic variation, and associating markers with traits in plant germplasm [37]. Unigenes derived microsatellite markers overcome the problem of redundancy in the EST database and have the advantage of assaying variation in the transcribed regions with their unique identities and positions [38]. The majority of SSRs in this study were dinucleotide repeats (38,848, 37.91%) based on the transcriptomic data of *P. polyphylla* var. *yunnanensis*. Another previous study of *P. polyphylla* var. *yunnanensis* transcriptome showed different conclusions that suggested that the monucleotide repeat type (56.3%) was the most abundant [39]. Nevertheless, distribution results in the present study are consistent with the frequencies of microsatellites among the gene indices of 24 plants in the previous study, which indicates that dinucleotide repeats are the majority of SSRs [40]. In addition, A/T repeat motif accounted for the largest proportion (91.74% of total mononucleotide repeats) in this study, which confirmed that the plant is rich in AT repeats [37]. The distribution density was estimated as one SSR per every 2.61 kb in this study. SSR loci in *Glycine* was proved to be three times more likely to occur in translated regions when derived from transcriptomic data than genomic data [41]. The density appears to vary significantly across plants through SSR density analyses based on transcriptomic data, i.e., *Populus wulianensis* (2.64 kb) [42], *Fagopyrum esculentum* (8.21 kb) [43], and arrowhead (9.13 kb) [44]. In contrast to microsatellite markers developed from genomic library, EST–SSRs can contribute to direct allele selection because they have known or putative functions and may be associated with the targeted trait [8]. Previous research has shown that SSRs have many important functions in terms of development, gene regulation, evolution, etc. [45]. The locations of SSRs appear to determine the types of functional role SSRs might play, and changes in SSRs

in different genetic locations can lead to changes in the phenotypes of an organism [46]. As polyphyllins are the main bioactive ingredients of the *Paris* species, this study developed EST-SSR markers based on the unigenes related to polyphyllin biosynthesis for the first time. SSRs scattered in the gene candidates were involved in the upstream and downstream of polyphyllin backbone biosynthesis. SSRs in coding regions can determine whether or not a gene gets activated or whether the protein product is truncated [46]. The most common SSR motif types related to polyphyllin (mononucleotide and dinucleotide) were in the accordance with those of the transcriptome.

### 4.2. Marker Polymorphism

A total of 10 SSR markers developed here are the first set of microsatellites related to the bioactive ingredient biosynthesis for *Paris* species. For all loci analyzed, the average *Ho* was obviously lower than the average *He*, indicating that self-pollination may be more common than is usually assumed in *P. polyphylla*; however, an excess of homozygotes may also result from sub-population (Wahlund effect). The average *He* is similar to that of the makers (0.5251) from isolating microsatellites from a (CT)n-enriched genomic library of *P. poyphylla* var. *chinensis* [47] and the average PIC is similar to that of random SSR makers (0.5355) from the root transcriptome of *P. polyphylla* [13]. As a whole, the average *Ne* (2.792), *He* (0.5600), and PIC (0.5225) showed the relatively high genetic polymorphism levels of these 10 loci [34]. Among the loci, locus 1150P7 had the lowest level of genetic diversity, whereas locus 1035P22 had the highest level of genetic diversity. The Shannon information of the loci showed that less information occurred among population, and more information occurred within population. In addition, locus 1035P9 and locus 1150P9 were in accordance with HWE, but the rest showed significant departures from HWE after Bonferroni correction, apparently due to heterozygote deficiency. HWE departures can be caused by intrinsic factors in the studied sample and by specific marker characteristics like mutation rates [48]. To clarify, the high number of loci deviating from HWE could partly be a result of sampling, the presence of null alleles, or might arise from selective pressure on the coding regions [49]. Null alleles can occur due to mutations in primer binding sites and lead to the overestimation of homozygosity [50]. Low levels of *Ho* here partly support the latter hypothesis. According to previous studies, there are approximately 30 microsatellites developed with small samples (10–60 samples) using a (CT)n-enriched genomic library, a magnetic bead enrichment strategy, or the transcriptome of a root [13,47,51]. However, all of these random markers without functional data were subsequently applied in 115 samples for a genetic diversity study of *P. polyphylla* var. *yunnanensis* and only 7 of them were validated to be efficient [52]. In this study, 10 markers of EST-SSR were derived from SSR related to the polyphyllin backbone biosynthesis; they were screened from 34 candidate markers after conducting experimental evaluation multiple times. Although the study was based on a limited number of markers, the results should be considered in future germplasm utilization and molecular-assisted breeding for *P. polyphylla*. The steady progress in microsatellite markers will benefit the genetic diversity and molecular breeding of *P. polyphylla* and ultimately help increase yields for this medicinal herb and other *Paris* plants.

### 4.3. Relationships in the Germplasm Diversity

To preserve the natural population and ensure a steady and renewable source of *P. polyphylla* for ethnomedical purposes, thriving cultivation of seedlings and planting has become essential in recent years [53]. The wild *P. polyphylla* and its varieties are rather rare; thus, populations collected in this study are cultivars of *P. polyphylla* var. *yunnanensis* from the representative growing areas, including the short-stalked variety and the long-stalked variety widely planted. The genetic diversity analysis of populations showed that populations QJS and LS had the higher genetic diversity than other populations, whereas population TC had the lowest genetic diversity. Population has a high level of genetic diversity, suggesting that it has strong capability to adapt to stressful environmental conditions [54]. The overall populations exhibited a low degree of differentiation among populations, but maintained a

high degree of genetic diversity among individuals, which were revealed by the results of AMOVA and genetic diversity estimation. A considerable degree of differentiation among individuals can be explained by cross-pollination and hybridization, since *P. polyphylla* is an insect-pollinated plant [55]. Although the high $F_{IS}$ of 5 loci suggested a high degree of inbreeding, the self-pollination rate is found to be low in the agricultural cultivation [1,36]. This situation is even more striking when the different cultivars are introduced and grown simultaneously in the plant base. The low $F_{ST}$ also implied a low level of differentiation among the populations. As the high average $Nm$ (>1.82) was detected, gene flow played a role in homogenization for populations and effectively suppressed the genetic differentiation that resulted from gene drift [56]. The considerably high gene flow might be indicative of an earlier period of more pronounced gene flow when the species had a more continuous distribution [57]. The genetic structure revealed that 10 populations probably derived from two ancestral groups and all germplasms were found to be have different levels of admixture. However, the two groups did not quite tally with the two cultivation groups at population level and the samples from the two cultivation groups from different populations were mixed with one another at the individual level. Moreover, cultivated populations also showed high genetic variation, consisting of the genetic diversity investigation of wild and cultivated populations of *P. polyphylla* using ISSR [10]. It can be speculated that they have originated from mixed provenances; thus, screening for superior provenances should be carried out as soon as possible [11]. Most populations of endangered species are commonly subdivided into different breeding groups, such as different breeds in the case of domestic plants, which are, in turn, subdivided into smaller reproductive units more or less interconnected [58]. Hence, the populations with higher genetic diversity have more utilization potential for resource conservation and selection of breeding materials.

## 5. Conclusions

In this study, a total of 10 EST-SSR makers related to polyphyllin backbone biosynthesis were developed based on the transcriptome of *P. polyphylla* var. *yunnanensis*. The novel SSR loci showed relatively high genetic polymorphism levels. The overall populations exhibited a low degree of differentiation among populations, but maintained abundant genetic diversity among individuals. The clustering groups of populations were different from cultivated groups, resulting in interspecific and intervarietal hybridization. The ten novel markers of EST-SSR provide an important tool for exploring the genetic diversity of *P. polyphylla*, and they will assist in developing efficient strategies for the germplasm resource management and conservation of this medicinal plant. The findings of this study may facilitate maker-assisted breeding and genetic engineering schemes involving this species, and other medicinal plants of the genus *Paris*.

**Supplementary Materials:** The following supporting information can be downloaded at: https://www.mdpi.com/article/10.3390/d14080589/s1, Figure S1: The distribution of six SSR motifs identified in the transcriptome. (a) number of mononucleotide repeats; (b) number of dinucleotide repeats; (c) number of trinucleotide repeats (display the top 50% of total number); (d) number of tetranucleotide repeats (display repeat number > 23); (e) number of pentanucleotide repeats (display repeat number > 10); (f) number of hexanucleotide repeats (display repeat number > 5). (c–f) only display the repeats with high occurrence frequency; Figure S2: The diversity and variation analysis within populations and among populations. (a) Shannon informational diversity statistics partitioned by population and total for codominant data (b) analysis of molecular variance using allelic distance matrix for F-statistics. Figure S3: PCoA of 136 individuals. The ten populations are denoted with different color.

**Author Contributions:** C.L. and X.G. brought the idea and managed the research. X.G., W.M. and B.Y. (Bin Yang) collected the samples. Q.S., B.Y. (Baolin Yao) and W.Y. implemented biological. X.G. conducted the data analysis. X.G. prepared the manuscript. All authors have read and agreed to the published version of the manuscript.

**Funding:** This research was funded by National Natural Science Foundation of China, grant number 31800273, 31970609; Yunnan Fundamental Research Projects, grant number 202001AT070114; Crop Varietal Improvement and Insect Pests Control by Nuclear Radiation; Startup Fund from Xishuangbanna Tropical Botanical Garden; 'Top Talents Program in Science and Technology' from Yunnan Province. Publication costs are funded by National Natural Science Foundation of China (Grant No. 31800273), Yunnan Fundamental Research Projects (Grant No. 202001AT070114), and Crop Varietal Improvement and Insect Pests Control by Nuclear Radiation.

**Institutional Review Board Statement:** Not applicable.

**Informed Consent Statement:** Not applicable.

**Data Availability Statement:** The data presented in this study are available in NCBI database with accession number PRJNA682903 and PRJNA630028.

**Acknowledgments:** We thank the following people for kind help in this study: Guoqiang Zhang (Shandong University) and Hanrui Bai (University of Science and Technology of China). We also thank the Central Laboratory of Public Technology Service Center at Xishuangbanna Tropical Botanical Garden, Chinese Academy of Sciences for providing the computer resources and technical support.

**Conflicts of Interest:** The authors declare no conflict of interest.

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
