# Peer review of "Development of EST-SSR Markers Related to Polyphyllin Biosynthesis Reveals Genetic Diversity and Population Structure in Paris polyphylla"

_diversity, doi:10.3390/d14080589_

Round 1

Reviewer 1 Report

Review of “Development of EST-SSR markers related to polyphyllin biosynthesis reveals genetic diversity and population structure in Paris polyphylla”. Based on the marker development and genetic diversity, this paper organized very well, provide insights into genetic diversity of the germplasm resources, and give useful information for facilitate marker-assisted breeding of this species. I think this paper can be accepted after minor revision.

There are some details bellow:

Line 68-70 Compared with SSR, expressed sequence tag simple sequences repeat (EST-SSR) has the advantage of more transferability among plant species, and is widely used in plant genetic mapping [8, 9].

Here the “is” can be delete.

Line 71 “relationship or” can be delete.

Line 76 genetic variations

Line 82 maybe change “and ” to “which”

Line 85 before “and” you need add a comma

Line 92-93 move “to” behind “were”

Line 109 delete “it was”

Line 118 delete “and”

Line 149 Ohtani et al. (2013) [21].

Line 156 before “and ” need a comma

Line 159 and gene flow

Line 162 change “and ” to “that”

Line 165 Here maybe need a reference

Line 166 maybe need change the first “and ” to “,”

Line 166-169 the sentence need rewrite to make it more fluency

Line 169 a comma before “and”

The consensus tree was generated, edited, and visualized using Phylip version 3.68, MEGA version 5.10, and FigTree version 1.4.2 respectively.

Line 172 here need the reference for STRUCTURE

Line 190 In Table 1, the “Pop” is better use the  complete spelling. The “N” and “E” need marked in the bracket.

 Line214-215 What is the mean of “mono”, “di”, “tri”, “tetra”, and so on. Here you need to give the complete spelling of these abbreviation under the figure legend. And the first letter of number need upper case.

Line 321-323 Here, the Figure 6 looks like repetition with Figure 5 in some ways. I suggest that put Figure 6 as supplementary materials.

Line 391 and Line 394 Paris need italic

Line 430 among populations

Line 430 10 markers of EST-SSR

Author Response

Review of “Development of EST-SSR markers related to polyphyllin biosynthesis reveals genetic diversity and population structure in Paris polyphylla”. Based on the marker development and genetic diversity, this paper organized very well, provide insights into genetic diversity of the germplasm resources, and give useful information for facilitate marker-assisted breeding of this species. I think this paper can be accepted after minor revision.

Response: Thank you very much for your positive comments and valuable suggestions. We have carefully improved the manuscript according to your suggestions.

There are some details bellow:

Line 68-70 Compared with SSR, expressed sequence tag simple sequences repeat (EST-SSR) has the advantage of more transferability among plant species, and is widely used in plant genetic mapping [8, 9].

Here the “is” can be delete.

Response: “is” has been deleted.

Line 71 “relationship or” can be delete.

Response: “relationship or” has been deleted.

Line 76 genetic variations

Response: The phrase has been changed into “genetic variations”.

Line 82 maybe change “and ” to “which”

Response: This sentence has been rephrased, and the “which” has been added here.

Line 85 before “and” you need add a comma

Response: “and” has been added here.

Line 92-93 move “to” behind “were”

Response: Thank you. We have moved the “to” forward.

Line 109 delete “it was”

Response: “it was” has been deleted.

Line 118 delete “and”

Response: “and” has been deleted.

Line 149 Ohtani et al. (2013) [21].

Response: Thank you. It is correct now.

Line 156 before “and ” need a comma

Response: A comma has been added before “and”.

Line 159 and gene flow

Response: “and” has been added here.

Line 162 change “and ” to “that”

Response: It has been changed.

Line 165 Here maybe need a reference

Response: Thank you very much. The reference of GenAlEx program has been cited.

Line 166 maybe need change the first “and ” to “,”

Response: The first “and” has been changed into a comma.

Line 166-169 the sentence need rewrite to make it more fluency

Response: This sentence has been rephrased for better understanding. It was updated as “The Nei’s (1983) standard genetic distance among populations, individuals, and the clustering trees based on the unweighted pair group method with arithmetic means (UPGMA algorithm (bootstrap: 1000) were calculated and analyzed using PowerMarker version 3.25”.

Line 169 a comma before “and”

Response: A comma has been added here.

The consensus tree was generated, edited, and visualized using Phylip version 3.68, MEGA version 5.10, and FigTree version 1.4.2 respectively.

Response: Thank you. This sentence has been updated.

Line 172 here need the reference for STRUCTURE

 Response: Thank you. The related reference has been added here.

Line 190 In Table 1, the “Pop” is better use the complete spelling. The “N” and “E” need marked in the bracket.

 Response: We totally agree with you. Table 1 has been updated in the revision.

Line214-215 What is the mean of “mono”, “di”, “tri”, “tetra”, and so on. Here you need to give the complete spelling of these abbreviation under the figure legend. And the first letter of number need upper case.

Response: Thank you very much. The figure and the legend have been revised according to your suggestion.

Line 321-323 Here, the Figure 6 looks like repetition with Figure 5 in some ways. I suggest that put Figure 6 as supplementary materials.

Response: Thank you. This figure has been transferred to the supplementary materials and cited as Figure S3 according to your suggestion.

Line 391 and Line 394 Paris need italic

Response: Thank you very much. They have been corrected, and the other species mentioned in the manuscript also were carefully checked to ensure correct format.

Line 430 among populations

Response: Thank you. It has been corrected.

Line 430 10 markers of EST-SSR

Response: Thank you. It has been revised.

Reviewer 2 Report

Paris polyphylla var. yunnanensis is only one of many varieties. Genetic diversity and population structure of this variety may not represent all Paris polyphylla. I suggest the description of the main results and title should be limited to Paris polyphylla var. yunnanensis. 

Paris polyphylla var. yunnanensis is widely distributed to Sichuan, Yunnan, Tibet and Guizhou. Is Yunnan province the main cultivated area of Paris polyphylla var. yunnanensis? As all populations were sampled from Yunnan province, it is necessary to explain why the sampling is limited to Yunnan. 

Figure 1, how to differentiate from long and short stalk population, It seems to be same height between Populaiton LS and TC in the west, but they are different varieties. Please define them in the text or lengend. 

In Results, content of 3.1 Sampling population distribution should be included into Materials and Method instead of Results.

Author Response

Paris polyphylla var. yunnanensis is only one of many varieties. Genetic diversity and population structure of this variety may not represent all Paris polyphylla. I suggest the description of the main results and title should be limited to Paris polyphylla var. yunnanensis

Paris polyphylla var. yunnanensis is widely distributed to Sichuan, Yunnan, Tibet and Guizhou. Is Yunnan province the main cultivated area of Paris polyphylla var. yunnanensis? As all populations were sampled from Yunnan province, it is necessary to explain why the sampling is limited to Yunnan. 

Response: Thank you very much for your suggestions. We chose P. polyphylla var. yunnanensis as the representative for P. polyphylla to explore and provide insights into the diversity of different germplasms of current cultivation areas. Firstly, P. polyphylla has seven varietas (Li 2008), but most of the varietas are rare in the wild and field. It is very hard to collect populations of either P. polyphylla or P. polyphylla var. polyphylla. Secondly, P. polyphylla var. yunnanensis is regarded as one of the important original species (Takhtajan, 1983). P. polyphylla var. yunnanensis accounts for a large proportion of Rhizoma Paridis yield for a long time, and it plays an important role in the traditional medicine trade (Cunningham, 2008). Thirdly, as P. polyphylla var. yunnanensis is mainly distributed and widely cultivated in Yunnan Province, the sampling was implemented in the major production areas in Yunnan. Therefore, we took the above factors into consideration when deciding the title.

Reference:

Cunningham, A. B.; Brinckmann, J. A.; Bi, Y. F.; Pei, S. J.; Schippmann, U.; Luo, P. Paris in the spring: A review of the trade, conservation and opportunities in the shift from wild harvest to cultivation of Paris polyphylla (Trilliaceae). J. Ethnopharmacol. 2018, 222, 208-216.

He, L. The Genus Paris Plants. Science Press: Beijing, China, 2008; pp. 33.

Takhtajan, A. A revision of Daiswa (Trilliaceae). Brittonia, 1983, 255-270.

Figure 1, how to differentiate from long and short stalk population, It seems to be same height between Populaiton LS and TC in the west, but they are different varieties. Please define them in the text or lengend. 

Response: The stalk length is one of the biggest differences between the long-stalk variety and the short-stalk variety, which is usually used as an important property for distinguishing the varieties in agriculture practice. The related description of differences in two varieties has been added in Materials and Methods section.

In Results, content of 3.1 Sampling population distribution should be included into Materials and Method instead of Results.

Response: According to your suggestion, the content of sampling has been totally transferred to the Materials and Methods section.

Reviewer 3 Report

The authors' article is quite complete, it can be called a complete and independent study that can be published in a journal. However, it is not clear to me how the study conducted by the authors correlates with the accumulation of polyphenols by individual representatives of the species? Is it known which individuals accumulated more phenolic compounds? How can these data be related to your research?

Author Response

The authors' article is quite complete, it can be called a complete and independent study that can be published in a journal. However, it is not clear to me how the study conducted by the authors correlates with the accumulation of polyphenols by individual representatives of the species? Is it known which individuals accumulated more phenolic compounds? How can these data be related to your research?

Response: Thank you very much for your positive comments. The bioactive compounds of P. polyphylla are polyphyllins. We investigated polyphyllin biosynthesis and accumulation, and identified the gene candidates related to polyphyllin biosynthesis in our previous study. Polyphyllin content of its rhizome is found to be different among cultivation areas. Based on the above background, we explored and revealed genetic diversity of the cultivated germplasms of two varieties based on SSRs derived from biosynthetic genes, assisting in the developing efficient strategies for the germplasm resource management of this medicinal plant. The present study mainly elucidates differentiation among populations based on genetic backgrounds involving in polyphylliln biosynthesis. Inspired by your comments, a large-scale of polyphyllin content detection of different cultivated populations will be implemented in our follow-up work. The correlation between polyphyllin content and population diversity, and the elite germplasms will be revealed and determined in the further study.